# α11β1 Integrin is Induced in a Subset of Cancer-Associated Fibroblasts in Desmoplastic Tumor Stroma and Mediates In Vitro Cell Migration

**DOI:** 10.3390/cancers11060765

**Published:** 2019-06-01

**Authors:** Cédric Zeltz, Jahedul Alam, Hengshuo Liu, Pugazendhi M. Erusappan, Heinz Hoschuetzky, Anders Molven, Himalaya Parajuli, Edna Cukierman, Daniela-Elena Costea, Ning Lu, Donald Gullberg

**Affiliations:** 1Department of Biomedicine and Centre for Cancer Biomarkers, University of Bergen, Jonas Lies vei 91, NO-5009 Bergen, Norway; cedric.zeltz@uhnresearch.ca (C.Z.); jahedul.alam@uib.no (J.A.); Hengshuo.Liu@helmholtz-muenchen.de (H.L.); p.m.erusappan@medisin.uio.no (P.M.E.); Himalaya.Parajuli@uib.no (H.P.); ning.lu@uib.no (N.L.); 2nanoTools Antikörpertechnik, Tscheulinstr. 21, 79331 Teningen, Germany; info@nanotools.de; 3Gade Laboratory for Pathology, Department of Clinical Medicine, University of Bergen, NO-5020 Bergen, Norway; Anders.Molven@uib.no; 4Department of Pathology, Haukeland University Hospital, NO-5020 Bergen, Norway; Daniela.Costea@uib.no; 5Cancer Biology Department, Fox Chase Cancer Center, Temple Health, Philadelphia, PA 19111, USA; Edna.Cukierman@fccc.edu; 6Department of Clinical Medicine, Center for Cancer Biomarkers CCBIO and Gade Laboratory for Pathology, University of Bergen, NO-5020 Bergen, Norway

**Keywords:** tumor microenvironment, tumor stroma, extracellular matrix, fibrillar collagen, cancer-associated fibroblasts, integrin alpha11

## Abstract

Integrin α11β1 is a collagen receptor that has been reported to be overexpressed in the stroma of non-small cell lung cancer (NSCLC) and of head and neck squamous cell carcinoma (HNSCC). In the current study, we further analyzed integrin α11 expression in 14 tumor types by screening a tumor tissue array while using mAb 203E3, a newly developed monoclonal antibody to human α11. Different degrees of expression of integrin α11 were observed in the stroma of breast, ovary, skin, lung, uterus, stomach, and pancreatic ductal adenocarcinoma (PDAC) tumors. Co-expression queries with the myofibroblastic cancer-associated fibroblast (myCAF) marker, alpha smooth muscle actin (αSMA), demonstrated a moderate level of α11^+^ in myCAFs associated with PDAC and HNSCC tumors, and a lack of α11 expression in additional stromal cells (i.e., cells positive for fibroblast-specific protein 1 (FSP1) and NG2). The new function-blocking α11 antibody, mAb 203E1, inhibited cell adhesion to collagen I, partially hindered fibroblast-mediated collagen remodeling and obstructed the three-dimensional (3D) migration rates of PDAC myCAFs. Our data demonstrate that integrin α11 is expressed in a subset of non-pericyte-derived CAFs in a range of cancers and suggest that α11β1 constitutes an important receptor for collagen remodeling and CAF migration in the tumor microenvironment (TME).

## 1. Introduction

The importance of the tumor microenvironment (TME) for the growth and spread of tumors is being increasingly recognized. In addition to serving as a structural scaffold, the extracellular matrix (ECM) serves as a reservoir of growth factors and cytokines that take part in the bidirectional communication between the stroma and the tumor cells [1,2]. The major cell types in the tumor stroma of solid tumors include cancer-associated fibroblasts (CAFs) of varying origin, endothelial cells, pericytes, mesenchymal stem cells, and immune cells [3,4]. CAFs represent a major cell type within the stroma contributing to ECM synthesis and ECM remodeling, and they also take part in the paracrine signaling, which affects the growth and invasive properties of the tumor cells, in chemoresistance and in the establishment of metastatic niches [3,4,5]. Importantly, a specific subset of myofibroblastic CAFs (myCAFs) has been implicated in the production of collagen [6]. CAFs produce collagen crosslinking enzymes of the lysyl oxidase (LOX) family, which increase the stiffness of the ECM and thereby affects the growth and invasion of tumor cells [7,8]. Fibroblastic cells thus constitute a group of mesenchymal cells of varying origins, some of which (i.e., myCAFs) share characteristics with the myofibroblasts that are found in granulation tissue during wound healing and tissue fibrosis [9].

In the context of pathological tissue and tumor fibrosis, the mesenchymally derived CAF population is thought to constitute a more heterogeneous cell mixture than the resident tissue fibroblasts in “resting” tissue. The balance among cells of different origins is dynamic in tissues showing tissue regeneration/fibrosis. In tissue fibrosis, genetically based cell linage tracing and a stringent use of antibodies have resulted in the characterization of activated fibroblasts that are derived either from endogenous fibroblasts [10,11,12], Gli+-positive mesenchymal stem cells (MSC) [13], or pericytes [14,15].

Pericytes exist as a major cell type in the pancreas and liver in the form of stellate cells [16,17], which proliferate and become activated in fibrosis models. The careful study from Öhlund et al. has defined a peritumoral alpha smooth muscle actin (αSMA)^high^ CAF population, termed myofibroblastic CAFs or myCAFs, which differ from a CAF population characterized by Il-6 production and referred to as inflammatory CAFs (iCAFs) [6].

The major sources of CAFs in tumors and tumor fibrosis are the endogenous tissue fibroblasts, pericytes, and ADAM12^+^ perivascular cells [15,18,19], and recently cell lineage tracing methods applied to transgenic polyoma middle T oncogene (PyMT) mice has somewhat surprisingly demonstrated a contribution from mesenchymal, non-hematopoietic bone marrow stromal cells to a PDGFRα-negative, clusterin-positive breast cancer CAF subpopulation [20].

Epithelial-mesenchymal transition (EMT) appears to be especially important in contributing to an invasive mesenchymal tumor cell type and creating niches for cancer stem cells [21], but these EMT processes in tumors have indirect consequences for the stroma. EMT has recently been studied in detail in *Lgr5CreER/Kras ^LSL − 12GD^/p53 ^fl/fl^*, genetic mouse model of squamous cell carcinoma (SCC) in which the tumors undergo spontaneous EMT [22]. These studies convincingly demonstrated that EMT occurs in a stepwise manner, which leads to the generation of subpopulations of tumor cells in different intermediate states between epithelial and mesenchymal. Interestingly, as the cells progressed towards EMT [22], the bona fide stroma changed in parallel, with regard to their composition, localization, and the presence of immune cells.

A detailed in vitro study using breast cancer cell spheroids identified a switch of tumor cells state into a mesenchymal invasive state without the tumor cells actually undergoing EMT [23]. The cells leading the way in this initial invasive migration, the “trailblazer cells”, were characterized by a mesenchymal seven-gene signature that was composed of *DOCK1*, *ITGA11*, *DAB2*, *PDGFRA*, *VASN*, *PPAP2B,* and *LPAR1* [23].

α11β1 integrin is a collagen-binding integrin that is expressed in mesenchymal cells identified as fibroblasts, myofibroblasts, and mesenchymal stem cells [24,25,26,27]. Relatively little is known about this protein in the context of tumors, but non-small cell lung cancers (NSCLC) and head and neck squamous cell carcinomas (HNSCC) express the α11 chain in activated stroma, where it has potential for serving as a biomarker for activated CAFs [28,29,30]. In the current report, we investigate the expression of integrin α11 chain in different tumor types and try to determine whether the expression of α11 subunit within a certain tumor type is able to mark a subpopulation of CAFs. We have generated and characterized an anti-human α11 mouse monoclonal antibody (mAb), mAb 203E3, for this purpose. In parallel, we have developed a function blocking α11 antibody, mAb 203E1, to test the functional involvement of α11β1 in collagen remodeling on CAFs.

## 2. Results

### 2.1. Generation and Characterization of Integrin α11-Specific Monoclonal Antibodies (mAbs)

Integrin α11 mAbs were generated at nanoTools, Germany (http://www.nanotools.de/), as described in Material and Methods, by immunizing mice with soluble human α11β1. Multiple-step screenings for binders of human α11β1 not cross-reactive with human α2β1 were performed while using the Luminex Assay and flow cytometry. The latter was used to select the clones that produced mAbs specific to human α11 while not recognizing human β1 or human α2 integrin chains. In this characterization, the previously described mouse C2C12 cell lines overexpressing human integrin α11, C2C12-huα11 (in C2C12-huα11 cells, human α11 chain heterodimerizes with mouse β1 integrin chain), and C2C12-huα2, were central [25]. To exclude cross-reactivity of the antibodies with the related α2 integrin chain, the mAbs were tested for reactivity with C2C12-huα2 cells, with no reactivity observed. To exclude reactivity with β1 integrin chain or other integrin α chains, hybridoma supernatants were screened against human A431 cells, which lack the expression of α11, but express human β1 chain and α2, α3, α5, and αv integrin chains [31]. In summary, no cross-reactivity with other integrins tested was noted. Two of the hybridoma clones producing mAbs 203E1 and 203E3, were further characterized and mAbs were affinity-purified. Both mAb 203E1 and mAb 203E3 caused a clear shift in the fluorescence intensity of the C2C12-huα11 cells in flow cytometry as compared with the non-expressing C2C12 cells (negative control; Figure 1a). The immunoprecipitation of α11 using mAbs 203E1 and 203E3, followed by Western-blotting with a polyclonal α11 antibody [32] confirmed the specificities of both antibodies for the 155 kD α11 band (Figure 1b), while the immunoctytostaining of C2C12-huα11 cells that were grown on collagen I showed the expected focal adhesion staining pattern (Figure 1c). Finally, the use of mAb 203E1 in cell attachment to collagen I and in cell spheroid migration assays in collagen gels demonstrated the effectiveness of mAb 203E1 in blocking α11-mediated adhesion both under two-dimensional (2D) and three-dimensional (3D) conditions (Figure 1d,e). In summary, the hybridoma clone 203E1 was identified as producing the blocking antibody mAb 203E1, while the clone 203E3 was identified as producing mAb 203E3 suitable for immunostaining. The immunoglobulin subtype and affinity determinations established that mAb 203E1 and mAb 203E3 are both of the IgG1 subtype (Hoschuetzky, H., nanoTools, Teningen, Germany, personal communication 2019), with affinities in the pM range (Appendix A).

### 2.2. Integrin α11 Expression in a Panel of Normal and Tumor Human Tissue Sections Using a Tissue Array

To screen for the expression of the α11 subunit in different tumor tissues, a tissue array with sections from 14 different tumor types and from corresponding normal tissues (Table 1) was screened using α11 mAb 203E3. The cytokeratin antibodies (anti-keratin 7 and 18) were used to distinguish epithelial/tumor cells from stromal cells. In agreement with previous studies in adult mouse tissues [33], the integrin α11 subunit levels in normal human tissues were low or below the detection limit in all of the normal tissues tested, except for the kidney specimen, where strong immunoreactivity was observed in the glomeruli, in a pattern that was compatible with positive mesangial cells (Figure 2, α11 expression in the normal kidney tissue section indicated by arrowheads).

Of the tumor tissues in the array, the breast, liver, lung, pancreas, ovary, and uterus tumors stood out as having markedly upregulated integrin α11. Co-staining with cytokeratin indicated the exclusive expression of α11 in the stroma. Weaker α11 expression was noted in the stroma of the small intestine and stomach adenocarcinomas. The skin squamous carcinoma section showed notable α11 chain expression, but it was restricted to a small region, which was perhaps due to the limited area of the section and the size spotted in the tissue array. Likewise, the immunostaining of the skeletal muscle rhabdomyosarcoma tissue was diffuse and will need to be confirmed in further sections. (Figure 2, integrin α11 expressions in the different tumor sections are indicated by arrows). In summary, four of the 14 tumor tissues analyzed lacked a specific integrin α11 signal, namely the stroma of the brain oligodendroglioma, the colon adenocarcinoma, the renal cell carcinoma and the prostate adenocarcinoma, whereas the majority of the carcinomas/adenocarcinomas showed upregulated integrin α11 expression in the stroma cells as compared with the normal tissues.

### 2.3. Characterization of Stromal Cells Expressing Integrin α11 Subunit in the Tissue Array Tumor Sections

Co-staining of the α11 subunit and cytokeratins were performed in combination with stromal markers to further characterize integrin α11 expression in the tissue array tumor sections. Three stromal markers were selected: fibroblast-specific protein 1 (FSP1, expressed in multiple cell types in the stroma, including immune cells [34]), αSMA (expressed in contractile activated fibroblasts, like myCAFs [6] and smooth muscle cells [35]), and vimentin (expressed in fibroblastic cells, endothelial cells, and pericytes [35]). We did not detect any α11 expression in the tumor cells (keratin-positive) of any of the sections tested, nor was the α11 chain detected in any of the larger blood vessels (αSMA-positive smooth muscle cells) using this limited set of markers. Co-localization of FSP1 with integrin α11 was only observed in breast and stomach adenocarcinoma sections, whereas αSMA and integrin α11 subunit co-localized to variable degrees in the stroma of most of the integrin α11-positive tumor tissues, which suggested that integrin α11 could be enriched in the cells corresponding to myCAFs. In the limited tissues pieces spotted on the arrays, no αSMA could be detected in either the skin sweat gland carcinoma or the stomach carcinoma (Figure 3). Finally, integrin α11 expression overlapped with vimentin expression in all sections tested, but most importantly, there were also integrin α11-positive, keratin-negative cells in the stroma with low or barely detectable vimentin expression, a phenotype that is compatible with these cells being CAFs.

To summarize this part of the investigation, the integrin α11 subunit immunostaining patterns of the tissue arrays suggest that, although α11 co-localization with FSP1, αSMA, and vimentin varies from one tumor to another, there is a trend for α11 and FSP1 to poorly co-localize in the tumor stroma, whereas α11 co-localized with αSMA to a larger extent in regions with activated stroma. Integrin α11 and vimentin also showed co-localization in the tumor stroma, but interestingly, α11 chain expression was also observed in stroma cells with no detectable expression of vimentin. A summary of the integrin α11 immunostaining and its co-localization with FSP1, αSMA, and vimentin in various tumor tissues, as seen in the screening results, is presented in Table 2.

### 2.4. RT-qPCR to Confirm the Immunostaining Data for the Tissue Arrays

To further verify the positive immunohistochemical data, we performed RT-qPCR to analyze levels of the integrin α11 subunit and the various markers in lung, pancreas, and skin on RNA isolated from the same tissues as used for preparing the tissue array sections. The RT-qPCR data demonstrated increased RNA levels of integrin α11 (ITGA11) in the lung, pancreas, and skin tumor tissue relative to the normal tissues, with the greatest increase in α11 RNA to be found in the pancreas tumor (Figure 4). Interestingly, vimentin (VIM) and αSMA (ACTA2) that we showed to co-localize with integrin α11 in the pancreatic cancer tumors also displayed increased expression in this tumor tissue.

### 2.5. Characterization of Cells Expressing Integrin α11 Subunit in PDAC and HNSCC

After screening the tumor tissue array, we examined the expression of α11 in the CAF subpopulations in the tumor stroma in more detail. Oncomine analyses of cancer datasets have identified ITGA11 overexpression in breast, pancreas, lung, colorectal-, and gastric cancer [36], and we have recently shown that HNSCC tumors express α11 in their stroma [30]. Based on these data, we decided to use tumor sections and isolated CAFs from PDAC and HNSCC tumors for further characterization of the expression and function of α11 using the novel α11 mAbs.

Six different stroma markers were chosen for co-staining with α11 mAb 203E3, while cytokeratin co-staining was performed to demarcate the tumor cells to better characterize the cells expressing α11. In agreement with previous data obtained with tissue sections from PDAC and HNSCC tumors, we observed α11 expression in the PDAC and HNSCC sections to be restricted to the stroma compartment and often seen peritumorally, in close in close contact with the tumor cells. The fibroblast-activating protein (FAP) staining was limited to the peritumoral region in close proximity to the tumor cells and it was extensively co-stained with α11. FSP1, on the other hand, showed little co-expression with α11, as was also the case with the pericyte marker NG2, which was expressed in distinct cell populations and appeared to not be co-expressed with α11 at all in the sections analyzed. The other two CAF markers, PDGFRβ and αSMA, were co-expressed with α11 in the majority of the stroma regions that were observed. Curiously, vimentin expression was again widespread in the stroma of both types of tumors, but it displayed a differential expression pattern from that of α11 (Figure 5 and Appendix A).

### 2.6. Role of Integrin α11β1 in Fibroblasts and CAFs

We screened a number of fibroblasts and CAFs for integrin α11 chain expression and its function in cell adhesive interactions with collagen I to examine the role of α11β1 as a collagen receptor in fibroblasts and CAFs (Figure 6). We also analyzed two additional collagen-binding integrin chains, integrin α1 (detected with the clone 639508 mAb) and integrin α2 (detected with P1E6 mAb and EPR 5788 mAb) chains, both dimerizing with the integrin β1 chain [37]. Human lung embryonic MRC5 fibroblasts expressed low levels of α11 in Western blotting by comparison to the α11-overexpressing C2C12 cells (C2C12-huα11, Figure 1), which served as a positive control for expression and functional analyses. The treatment with TGF-β resulted in a moderate increase in α11 levels in the MRC5 cells (Figure 6a and Appendix A). In agreement with this, the effect of 203E1 in these cells was restricted to cell adhesion, whereas collagen gel contraction and spheroid invasion were unaffected by the presence of mAb 203E1, which was presumably due to involvement of other collagen-binding integrins (Figure 6b). In agreement with the concept that the degree of inhibition is related to the degree of expression, the effect of 203 E1, and the combination of the function-blocking antibodies 203E1(α11) and P1E6 (α2) was also greater in all three functional assays (cell attachment, collagen gel contraction, and spheroid invasion) for human gingival fibroblasts (hGF) with higher α11 expression (Figure 6a,c).

Normal fibroblasts and CAFs from HNSCC tumors were analyzed afterwards. Here, the expression of α11 was modest in these CAFs (Figure 6a). The PDAC tumors, in turn, gave us access to two kinds of CAFs: control PDAC CAFs (P CAFs) and CAFs, in which integrin α5 had been knocked down using CRISPER-CAS-9 (P CAF KDα5) [38]. The latter CAFs expressed α11 at a comparable level to that achieved by hGF cells (Figure 6a), but the 203 E1 antibody in these cells was mainly effective in the invasion assay, with a smaller effect of mAb being observable in the cell attachment and collagen gel contraction assays. The less efficient inhibition of 203E1 was probably partly due the to higher levels of α2 integrin in P CAF KDα5 than in the hGF cells (Figure 6c,d), but it might also reflect some involvement of other β1 integrins in the adhesion of these cells to collagen I (see Discussion).

In summary, the function-blocking assays demonstrate that α11 mAb 203E1 inhibited cell-collagen interactions in a manner that seemed to depend on the level of α11 expression, as well as on the presence of other collagen receptors. MAb 203E1 was especially efficient in blocking 3D spheroid migration in P CAF KDα5.

## 3. Discussion

Most integrins are widely expressed on multiple cell types. Among the collagen-binding integrins, the α1 and α2 integrin chains are both expressed on dermal fibroblasts [39,40], but only limited data are available regarding their expression and their function in CAFs. However, in general, the α1 and α2 integrin chains are widely expressed and they can also be detected in tumor cells as well as vascular and immune cells [41,42,43]. These integrins are therefore not particularly useful as biomarkers for CAFs. The collagen-binding α10 integrin is normally limited to cartilage and a very restricted subset of fibroblasts [44]. Although melanoma cells have been reported to express α10, no expression has been reported in CAFs [45]. In contrast, α11β1 integrin is expressed in a pattern that is restricted to CAFs in NSCLC and HNSCC, as observed by immunostaining while using a polyclonal antibody to α11 [28,30,46].

Data from mouse models of NSCLC show that integrin α11 expression is associated with increased stiffness of the tumors, which suggested the involvement of α11β1-mediated ECM reorganization as an underlying mechanism and resulted in stiffer and more ECM [29]. In addition to the suggested direct effect of α11β1 in mediating collagen reorganization, a correlation with lysyl oxidase-like 1 (LOXL1) expression has been noted [29,47]. This indirect mode of regulating the levels of collagen cross-linking enzymes needs further studies to directly link it to an α11β1-mediated molecular mechanism.

The current work adds to existing studies of integrin α11 expression in various human tumors. In the process of adult tissue immunostaining using the polyclonal integrin α11 antibody, we noted problems with non-specific background staining, so that it was essential to develop new and better reagents. The monoclonal antibodies mAbs 203E1 (function-blocking) and 203E3 (immunostaining) described herein are both high affinity mono-specific mouse antibodies to the human integrin α11 chain with no reactivity with either the β1 integrin chain or the human α2 integrin chain, or any other tested integrin chains.

The α11 immunoreactivity in the stroma of invasive ductal breast cancer is interesting, since breast cancer tissue is often stiff and desmoplastic [48,49,50], which is in agreement with our current picture of integrin α11 expression as being enriched at sites of high mechanical stress [25]. The high integrin α11 protein expression that is seen in the invasive ductal mammary carcinoma data is supported by large-scale cancer genomics data at TCGA demonstrating high α11 mRNA expression in invasive breast cancer (TCGA Research Network: http://cancergenome.nih.gov/). Similarly, the analysis of an Oncomine database (https://www.oncomine.org/) supports the expression of α11 in various forms of breast cancer. A functional role for α11 in breast cancer is likewise supported by data from a PyMT mouse model, in which the absence of α11 in the breast cancer stroma greatly attenuates breast tumorigenesis and metastasis [36]. In this context, it is also interesting to note that human breast cancer tumor cells at the invasive front in an in vitro spheroid metastasis model express integrin α11 RNA at the point in time when the cells assume a mesenchymal invasive phenotype. Integrin α11 is part of the gene signature in these “trailblazer” breast cancer cells, and it is thought to be functionally involved in this invasion process [23]. Nevertheless, we could not observe any integrin α11 staining in the mammary cancer cells of the limited number of sections that we analyzed here.

The in vitro data that were obtained here with α11 function-blocking antibodies suggest that α11β1 has a role in CAF-mediated collagen remodeling and cell migration. Although the α11 function- blocking antibody almost completely inhibited cell-collagen adhesion interactions in α11-transfected C2C12 cells, the contribution of α11β1 to cell-collagen interactions was lower in the fibroblasts and CAFs expressing additional collagen receptors, including α2β1. Interestingly, the effect of antibodies to β1 integrin was still greater than the combined effect of the α2β1 and α11β1 blocking antibodies, which suggested the involvement of other β1 integrins in indirect cell adhesive interactions with collagen [51,52]. Integrin α5β1 is one candidate receptor for mediating these indirect interactions with the collagen matrix, since fibronectin is present during the collagen gel contraction and spheroid migration.

The strong immunoreactivity of integrin α11 protein in pancreatic carcinoma and ovarian cyst adenocarcinoma tissues is also in agreement with the TCGA expression data, where *ITGA11* expression in pancreatic cancer belongs to the top-five tumor category for all tumor types analyzed for α11 mRNA expression (TCGA Research Network: http://cancergenome.nih.gov/). A recent study of pancreatic cancer CAFs has suggested that α11 may play a role in cell migration on pancreatic CAFs [53]. However, it is important to point out that α11 has a tendency to be induced in cell culture. In our own work, we have failed to detect α11 integrin in breast cancer cells (this study and [36]) or in liver or pancreatic stellate cells [26,33], whereas work that was performed using in vitro cultured cells has suggested a role for stellate cell-derived α11-expressing CAFs in tissue and tumor fibrosis [53,54,55].

The lack of proper antibody controls is one weakness in some studies of integrin α11 using polyclonal antibodies. The use of polyclonal antibodies with pathological tissue sections is challenging. The monoclonal antibodies that were characterized here were generated using a soluble heterodimeric protein, which was immunized in native form, and the mAbs were screened with α11-specific reagents. No reactivity was noted with any of the other integrin α or β chains tested, and they should constitute a valuable tool for future work. Although the prostate tumor tissue that was analyzed here was negative for integrin α11 immunoreactivity, we have previously noted α11 expression in the prostate carcinoma stroma using the polyclonal integrin α11 antibody [56]. The new data agree with expression data that are available from TCGA, in which the α11 mRNA levels reported in prostate adenocarcinoma are modest. The α11 immunostaining observed in cells expressing low levels of vimentin is interesting. Vimentin has been widely regarded as a universal marker of stromal cells [4], but curiously resident mesenchymal stem cells (MSCs) have recently been reported to be characterized by a low expression of vimentin [57].

When activated, fibroblasts become contractile and they produce and remodel collagen. During the activation process, normal quiescent fibroblasts first become protomyofibroblasts and then, when fully activated, are known as myofibroblasts [58,59]. One marker myofibroblasts is αSMA. It is worth noting that data are now accumulating to suggest that αSMA is an inconsistent marker of activated collagen-producing myofibroblasts cells, at least in fibrotic conditions in the lung, kidney, and heart [60,61]. Independent in vitro data on activated fibroblasts, in tumors known as myCAFs [6], share characteristics with the CAFs that are described in this study with integrin α11 chain expression associated with αSMA expression and a myofibroblast phenotype [27,29,62,63,64,65]. Interestingly, we found here that CAFs expressing integrin α11 do not systematically co-express αSMA, since we noted a strong co-expression of integrin α11 and αSMA in CAFS around the tumor cells of the PDAC and HNSCC sections, which suggested that these α11^+^-CAFs could have a role in collagen remodeling at the border of the tumor in order to facilitate tumor cell invasion. In colon cancer, the role of tumor cell αvβ3 at the tumor cell-stromal cells interfaces has been shown to be intimately connected with CAF osteopontin expression and the formation or generation of a cancer stem cell niche [66]. Recent studies using six antibody markers actually classified four different subtypes of CAFs in breast cancer TME, where the peritumoral CAFs expressed αSMA and FAP and they were found to be immunosuppressive [67]. As already mentioned, similar studies of PDAC tumors have identified a myofibroblastic CAF subtype, myCAF, at the tumor stroma interfaces, and an inflammatory subtype iCAF at a greater distance away from the tumor cells [6].

Thus, the data that are presented here raise a number of interesting questions. One central issue concerns the origin of integrin α11-expressing CAFs in the tumor stroma and whether these have a common developmental origin. During development, integrin α11 is highly expressed in the neural crest-derived head mesenchyme, in addition to the mesenchyme contributing fibroblasts to tendons, periosteoum, and perichondrium, but also in αSMA-positive myofibroblasts in the intestinal villus cluster [25]. Villus cluster myofibroblasts are thus naturally occurring myofibroblasts. Here, we also identified certain α11-expressing cells in the kidney mesangium that are α11 positive. We suggest that these cells represent the mesangial myofibroblasts, but this will require further characterization work. In the PyMT mouse model of breast cancer, some CAFs have been shown to originate from the bone marrow (BM) compartment [20]. Once these cells from the BM have arrived in the breast cancer TME, they expand and differentiate into CAFs. Interestingly, this subset of CAFs lack PDGFαR. It will be interesting to determine the origin of the α11-expressing CAFs in breast and pancreatic cancer, especially in the light of data demonstrating the expression of α11 in a subset of mesenchymal stem cells [68].

Secondly, it will be interesting to determine which factors drive integrin α11 expression. Based on our current knowledge, it is tempting to speculate that the stiffness of the tumor tissue will be one factor, which raises the α11 expression levels via unknown mechanisms. Furthermore, the finding that integrin α11 expression is high in desmoplastic tumors raises the question of how α11β1 on CAFs contribute to collagen synthesis.

Finally, our immunohistostaining data with a limited set of markers clearly demonstrate that CAFs in the stroma are heterogeneous with regard to α11 and αSMA expression, which suggests that α11 is expressed on distinct subset(s) of CAFs. In the light of xenograft models, existing data suggest that some CAFs expressing α11β1 are tumor supportive [28,29], and future studies should be aimed at better defining the α11-expressing CAF subsets in various tumor types, including the α11^+^/vim^high^ and α11^+^/vim^low^ subsets.

## 4. Materials and Methods

### 4.1. Tissue Array Sections

Frozen Tumor and Normal Tissue Array sections from BioChain Institute Inc. (Newark, CA, USA, Cat# T6235700-5, Lot#B712100, five sections per array) were used to examine α11 expression in human normal and tumor tissues. Immunostaining was also performed on fresh-frozen tumor tissue sections from patients that were diagnosed with pancreatic ductal adenocarcinoma (PDAC) or head and neck squamous carcinoma (HNSCC), which were both obtained from Haukeland University Hospital and subject to ethical approval from the Committee for Ethics in Health Research of West Norway (permit numbers REK Vest 2013/1772 and 2010/481, respectively).

### 4.2. Cells and Reagents

The C2C12 cells stably expressing human α11 integrin or human α2 integrin subunits (C2C12-huα11 and C2C12-huα2, respectively) have been described previously [25]. MRC5 human lung fibroblasts (American Type Culture Collection) were obtained from Robert Lafyatis laboratory (University of Pittsburgh Medical Center, Pittsburgh, PA, USA), the primary hGFs were isolated from healthy gingival tissue, as described earlier [69], and the primary oral cancer-associated fibroblasts (CAFs) and the primary normal oral fibroblasts (NOFs) were isolated from the same patient that was diagnosed with HNSCC at Haukeland University Hospital. The pancreatic cancer CAFs and integrin α5 knockdown CAFs isolated from PDAC, as described in [38], were obtained from Edna Cukierman’s laboratory (Fox Chase Cancer Center, Philadelphia, PA, USA). All of the cells were attested as mycoplasma-free using the Lonza Mycoalert mycoplasma detection kit (Fisher scientific, Gothenburg, Sweden, Cat# 11630271) and they were cultured in DMEM with GlutaMAX (Gibco, Life technology limited, Paisley, PA49RF, UK, Cat# 31966-021) supplemented with 10% fetal bovine serum (Gibco, Life technology limited, Cat# 10270-106) and 1% Penicillin-Streptomycin (Sigma, St Louis, MO, USA, Cat# P4333). TGF-β1 was from PeproTech (Hamburg, Germany, Cat# 100-21C).

### 4.3. Generation of Mouse Monoclonal Antibodies Specific to the Human Integrin α11 Chain

The integrin α11 mAbs were custom-made at nanoTools (http://www.nanotools.de/) while using established procedures. Briefly, NT-HRM mice (nanoTools Antikoerpertechnik, Germany) were immunized with soluble recombinant human α11β1 integrin protein produced in CHO cells (R&D Systems, Minneapolis, MN, USA, Cat# 6357-AB), boosted twice, and cell fusion performed on day 68. Fusion was performed from 12 mice and hybridomas were screened for α11-producing clones in several steps. Luminex beads that were coated with α11β1 integrin were used to screen the α11 binders. Supernatants from positive clones were tested in flow cytometry for a positive signal with C2C12-huα11 cells [25], but a lack of reactivity with cells not expressing human α11 (parental mouse C2C12 cells and A431 cells, which express human β1 integrin, together with a number of other human integrin α chains, including αv, α2, α3, and α5 [31]). The positive supernatants were tested for their ability to immunostain focal contacts α11- containing in C2C12-huα11 cells that were plated on collagen I and to inhibit cell attachment of C2C12-huα11 cells to collagen I, but not to fibronectin. Limited dilution further characterized and finally subcloned positive clones.

### 4.4. Flow Cytometry

The C2C12-huα11 cells were detached and neutralized with DMEM with FBS. After being washed three times with PBS (without Ca^2+^ and Mg^2+^), they were blocked with 5% BSA for 30 min at room temperature (RT). They were then mixed with mAb 203E1 or mAb 203E3 (3 µg/mL each) and then incubated for 1 h at 37 °C, followed by washing three times with PBS and incubation for 1 h in RT with Alexa fluor^®^ 647-conjugated goat anti-mouse IgG (1:400, Jackson ImmunoResearch, Cambridgeshire, UK). Finally, the cells were washed and analyzed by FACS Accuri at the Molecular Imaging Center (MIC, University of Bergen, Bergen, Norway). FLOWJO computer software was used for data analysis (FLOWJO, LLC, Franklin Lakes, NJ, USA).

### 4.5. Immunoprecipitation

Subconfluent C2C12-huα11 cells were cultured in 10 cm Petri dishes and lysed in 1 mL lysis buffer (10 mM Tris-HCl pH 7.4, 150 mM NaCl, 0.5% NP40, 1 mM MgCl_2_, 1 mM CaCl_2_, and complete Mini, EDTA-free cocktail (Roche Diagnostics GmbH, Manheim, Germany, Cat# 11836170001) for 20 min at 4 °C on a rocker. Protein lysates were centrifuged at 13,000× *g* for 20 min at 4 °C. The supernatants were incubated with 50 μL of protein G Sepharose beads (GE Healthcare, Uppsala, Sweden, Cat# 17-0618-01) with control non-immune mouse IgG for 2 h in a rotator at 4 °C, and spun down at 5000 rpm for 2 min at 4 °C. The resulting supernatants were then collected and incubated with 5 μg/mL of primary antibody (rabbit polyclonal anti-human α11 antibody or mAbs 203E1 or 203E3) overnight at 4 °C. The samples were incubated with 50 μL of protein G Sepharose beads for 2 h at 4 °C and spun down at 5000 rpm for 2 min at 4 °C. The beads were washed twice in PBS and 50 μL of 2× sample buffer with reducing agent was added before the boiling samples for 5 min. Finally, the samples were centrifuged for 2 min at 5000 rpm and loaded onto 6% SDS-PAGE gels for the separation of proteins, which were transferred to PVDF membranes while using the iBlot^®^ system (Invitrogen, Kyrat Shmona, Israel, Cat# IB301002). The immunoprecipitated proteins were detected by incubating the membranes with polyclonal α11 rabbit antibody [32] followed by goat anti-rabbit HRP (see Western blotting for details).

### 4.6. Western Blotting

The cells cultured in monolayers were washed with phosphate-buffered solution (PBS, Sigma-Aldrich, St Louis, MO, USA) lysed in SDS-sample buffer (Bio-Rad, Oslo, Norway, Cat# 1610791) with 3% of 2-β-mercaptoethanol (Sigma-Aldrich, Cat# M7154) and sonicated using a Vibra-Cell™ ultrasonic processor (Sonics and Materials, Newtown, CT, USA). The cell lysates were subjected to (6% acrylamide) SDS-PAGE electrophoresis after boiling for 5 min., and the proteins were transferred to PVDF membranes using the iBlot^®^ system. The membranes were blocked with 5% non-fat dry milk (Marvel, UK) in Tris-buffered saline containing 0.1% Tween20 (TBS-T), incubated with primary mouse anti-human α11 antibody Mab 210F4 [70] or rabbit monoclonal anti-human α2 (EPR 5788, Abcam, Cambridge, MA, USA, Cat# ab133557) or mouse monoclonal anti-human α1 antibody (R&D Systems, Minneapolis, MN, USA, Cat# MAB 5676) and anti-β-actin (AC-74, Sigma-Aldrich, Cat# A5441) overnight at 4 °C. Following the incubations, the membranes were washed in TBS-T three times for 10 min and incubated with goat anti-mouse- or goat anti-rabbit-HRP-conjugated secondary antibodies for 1 h at room temperature. The membranes were developed while using the ECL™ western blotting systems kit (GE Healthcare) and photographed using the ChemiDoc XRS device and the Quantity One 1-D Analysis Software (Bio-Rad).

### 4.7. Immunocytofluorescence

C2C12 and C2C12-huα11 cells were seeded on coverslips that were pre-coated with bovine collagen I (Advanced BioMatrix, PureCol, Carlsbad, CA, USA, Cat# 5005) and cultured for 4 h. The coverslip coating was done in a 24-well plate with collagen I solution at a final concentration of 100 μg/mL, followed by incubation overnight at 4 °C. After culturing, the cells were briefly washed with PBS and fixed in 4% PFA for 10 min., washed in PBS (3 × 5 min), permeabilized, and blocked with 0.1% TritonX-100 and 1% BSA in PBS at RT for 1 h. For integrin α11 detection, the cells were incubated with affinity-purified α11 mAb, either mAb 203E1 or mAb 203E3 (0.32 mg/mL and 0.5 mg/mL, respectively, both diluted 1:200). The antibodies were diluted in 10% goat serum in PBS and supplied on coverslips in a 24-well plate, 200 μL/well. After incubation at 37 °C for 1 h, the cells were rinsed in PBS/Tween-20 (three washes 5 min each). The secondary antibody was Alexa Fluor^®^ 488 AffiniPure goat anti-mouse IgG (Jackson ImmunoResearch, Cat# 115-545-062, 1:800) and TRITC-conjugated phalloidin (Sigma, St Louis, MO, USA, Cat# P1951, 1:100) was used to counter-stain stress fiber-associated actin. Both the secondary antibody and pahalloidin were diluted in PBS and applied to coverslips for 1 h at RT. The cells were rinsed for 3 × 5 min in PBS/Tween-20 and stained for 2 min with DAPI Nucleic Acid Stain (Molecular Probes). The staining results were recorded using a Zeiss Axioscope microscope (Zeiss, Oberkochen, Germany) that was equipped with an AxioCam camera (Zeiss) and Axiovision software (Zeiss).

### 4.8. Immunohistostaining

The tissue array sections or fresh tumor cryosections were fixed with methanol for 8 min at −20 °C, followed by rehydration in PBS (3 × 10 min). The unspecific binding sites were blocked using 10% goat serum in PBS and the sections were incubated with primary antibody combinations, as indicated in the figures. The primary antibodies used were: mouse anti-integrin α11 mAb (mAb 203E3, 0.5 mg/mL, 1:200), rabbit anti-human cytokeratin 7 mAb (R17-S, Novusbio, Centennial, CO, USA, Cat# NBP1-30152, 1:200), rabbit anti-human cytokeratin 18 mAb (Epitomics, Burlingame, CA, USA, Cat# 1433-1, 1:400), rabbit anti-human FAP mAb (My Biosource, San Diego, CA, USA, Cat# MBS33414, 1:200), rabbit anti-FSP1 pAb (Millipore, Darmstadt, Germany, Cat# 07-2274, 1:300), rabbit anti-NG2 pAb (Millipore, Cat# AB5320), and mouse anti-αSMA FITC-conjugated mAb (1A4, Sigma, Cat# F3777, 1:400). All of the primary antibodies were diluted in 10% goat serum in PBS. After incubation at 37 °C for 1 h, the slides were rinsed in PBS/Tween-20 (three times for 5 min). The secondary antibodies Alexa Fluor^®^ 594 AffiniPure goat anti-mouse IgG (Jackson ImmunoResearch, Cat# 115-585-062, 1:800) and Alexa Fluor^®^ 488 AffiniPure goat anti-rabbit IgG (Jackson ImmunoResearch, Cat# 111-545-045, 1:800) were diluted in PBS, applied to the sections, and incubated for 1 h at room temperature. The slides were then rinsed in PBS/Tween-20, the stained sections mounted in ProLong™ Gold Antifade Mountant with DAPI (ThermoFisher, Eugene, OR, USA, Cat# P36931). The staining results were recorded using a Zeiss Axioscope microscope that was equipped with an AxioCam camera and Axiovision software.

### 4.9. Quantitative Reverse Transcription Polymerase Chain Reaction (RT-qPCR)

The RT-qPCR was performed, as previously described [27]. One microgram RNA was used along with MMLV-derived reverse transcriptase (Bio-Rad) and a blend of oligo (dT) and random hexamer primers. Next, 20 ng of reverse-transcribed cDNA was used as a template, along with 0.5 µM of each primer, in a 20 µl qPCR reaction using FastStar Universal SYBR Green Master (Roche Applied Science, Penzberg, Germany), which was in accordance with the manufacturer’s protocol. RT-qPCR was performed in a Light-Cycler 480 Instrument II (Roche Applied Science). The qPCRs were performed in triplicate for each cDNA sample and negative controls where no cDNA template was included for each pair of primers. Table 3 lists the primers used for qPCR.

### 4.10. Cell Adhesion Assay

48-well plates were coated with human plasma fibronectin (2 μg/mL: Sigma-Aldrich, Cat# F0895) or bovine collagen type I (0,5 μg/mL: Bovine PureCol^®^, Advanced BioMatrix, Carlsbad, CA, USA, Cat# 5005) and incubated for 2 h at 37 °C. After washing the coated plates twice with PBS, they were blocked with 2% BSA for 1 h at 37 °C, and the cells were washed twice with DMEM without FBS. 1 × 10^5^ cells/well were incubated for 45 min at 37 °C with clone 11,711 (10 μg/mL, mouse IgG1 isotype control, R&D Systems, MN, USA, Cat# MAB002), mAb 203E1 (10 μg/mL, integrin α11 antibody), P1E6 (5 μg/mL, integrin α2 antibody, Merck Millipore, Cat# MAB1950Z), and mAb 13 (5 μg/mL, integrin β1 antibody, BD Biosciences, San Jose, CA, USA, Cat# 552828). Following incubation, the non-adherent cells were carefully removed by washing twice with PBS containing Ca^2+^ (1 mM) and Mg^2+^ (0.5 mM). The cells were fixed with absolute ethanol for 10 min at room temperature, washed twice with distilled water, and stained with 0.1% crystal violet for 25 min at room temperature. The plates were washed three times with distilled water and the cells were lysed with 1% Triton X-100/PBS for 15 min. The lysates were transferred to a 96-well plate and the absorbance was read at 595 nm (Spectramax^®^ Plus 384, Molecular Devices, San Jose, CA, USA).

### 4.11. Collagen Gel Contraction

Collagen gel contraction was performed according to a previously described protocol [69]. 24-well plates were blocked with 2% BSA overnight at 37 °C and washed three times with PBS. A collagen solution was prepared by mixing 50% of DMEM 2× (SLM-202-B, Merck Millipore, Cat# SLM-202-B), 10% of 0.2M HEPES at pH 8.0, and 40% of collagen type I. The solution was then mixed with cells to obtain a final concentration of 1 × 10^5^ cells/mL. 400 μL of cell-collagen suspension was added to each well and allowed to polymerize for 90 min at 37 °C. Antibodies were added to DMEM containing 0.5% FBS for the blocking experiments. Polymerized collagen gels were floated with 400 μL of DMEM. The gel diameters were measured using a ruler and percentage of the initial gel area was calculated at different time points.

### 4.12. Spheroid Preparation and Migration Assay in 3D Collagen Gel

Single cell type spheroids (homospheroids) were prepared by the hanging drop method, as described earlier [63]. Cells with 80% confluency were trypsinized and resuspended in a solution that was composed of ¾ volume of DMEM with 10% FBS and ¼ volume of methylcellulose (Sigma-Aldrich) to a concentration of 1 × 10^6^/mL. Approximately 35 drops (25 μL/drop, 2.5 × 10^4^ cells) were placed on the lid of a Petri dish containing DMEM in the bottom. The lid was inverted over the bottom of the dish. The spheroids were cultured for one day under regular cell culture conditions (37 °C and 5% CO2). A collagen solution was prepared by mixing 50% of DMEM 2×, 10% of 0.2 M HEPES at pH 8.0 and 40% of collagen type I, and 100 μL of this solution was added onto a 96-well plate and incubated for 15 min at 37 °C. 1 spheroid was embedded per well and the collagen-spheroid solution was allowed to polymerize for 90 min at 37 °C. After polymerization, 100 μL of DMEM with antibody was added to each well to cause the collagen gel to float, before culturing for 24 h. The spheroids were examined under an inverted light microscope (Leica DMIL, Wetzlar, Germany) and photographed. The resulting images were then analyzed and processed with Fiji. The modification of the spheroids included the alteration of the type to 8-bit, adjustment of the brightness/contrast, subtraction of the background, and establishment of a threshold. The radial Profile plugin of Fiji was applied to quantify the intensity of the cells.

## 5. Conclusions

In summary, our data indicates that integrin α11 is induced in CAFs in the stroma of tumor tissues that are characterized by high tissue stiffness and desmoplasia, and the morphology of the cells in the set of tumors analyzed here suggests a reactive stromal phenotype, which is not associated with vascular structures. Functional assays with cultured fibroblasts and CAFs demonstrated a role for α11β1 in collagen reorganization and CAF invasion, lending further support to the hypothesis that α11 might be an interesting candidate for stromal-targeted therapy to increase the efficacy of immune therapy, as well as conventional therapeutic approaches.

## Figures and Tables

**Figure 1 cancers-11-00765-f001:**
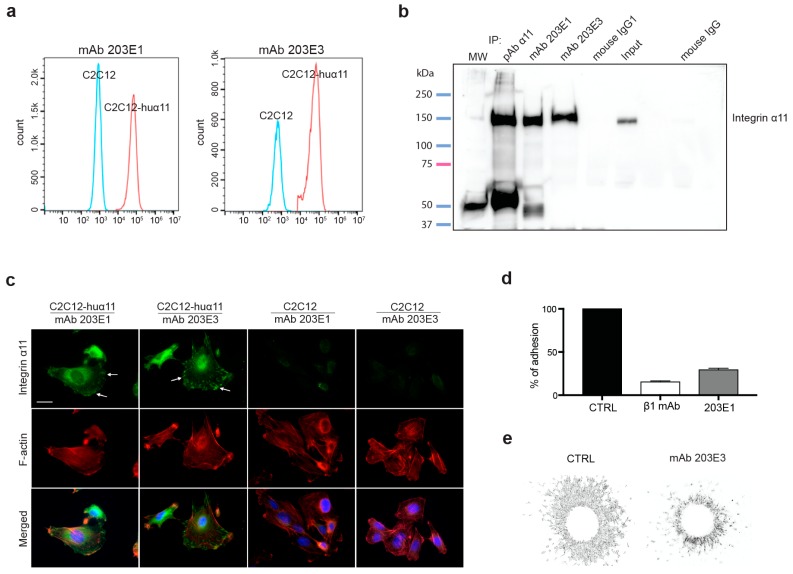
Characterization of the α11 mAb 203E1 and the mAb 203E3. (**a**) Characterization by flow cytometry. C2C12 cells and C2C12 expressing human α11 integrin (C2C12-huα11) cells were subjected to flow cytometry using the 203E1 and 203E3 mAbs. Only the C2C12-huα11 cells incubated with the 203E1 and 203E3 mAbs displayed a fluorescence shift. (**b**) Characterization by immunoprecipitation. Integrin α11β1 was immunoprecipitated with the 203E1 and 203E3 mAbs. The polyclonal α11 antibody (pAb α11) was used as a positive control, whereas mouse IgG1 was used as a negative control. Loading of the cell lysis (input) has been included to appreciate the efficiency of the immunoprecipitation. Immunoprecipitated proteins were detected with a rabbit polyclonal antibody to human α11. The full-size Western blotting is presented, MW: molecular weight marker. (**c**) Characterization by immunocytochemistry. C2C12 and C2C12-huα11 cells were plated on collagen I and immunostained using mAbs 203E1 and 203E3. Both antibodies immunostained focal adhesions (arrows). Scale bar: 20 µm. (**d**) Characterization in cell adhesion assay. C2C12-huα11 cells were incubated with either β1 mAb or 203E1 mAb and allowed to adhere to collagen I. (**e**) Characterization in invasion assay. Homospheroids composed of C2C12-huα11 cells were embedded in collagen I gel and treated with either the mouse IgG1 isotype control or 203E1 mAb at 10 µg/mL. Spheroid migration was analyzed after 24 h.

**Figure 2 cancers-11-00765-f002:**
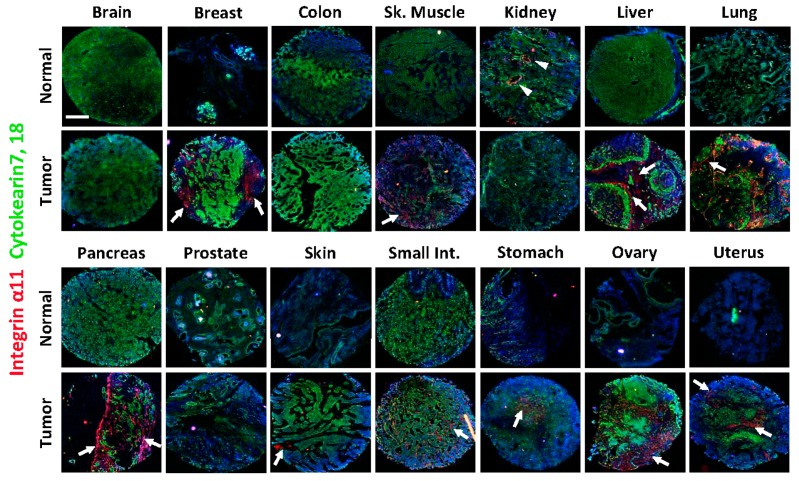
Expression of integrin α11 in array sections from normal and tumor adult human tissues. Immunofluorescence staining was performed on the sections using α11 203E3 mAb (red) and cytokeratin 7 and 18 (green). The cell nuclei were stained with DAPI (blue). In the normal tissues integrin α11 expression was only detectable in the kidney section (arrowheads), while in the tumor tissues α11 expression was detected in 10 of the 14 tumors tested (indicated by arrows in the respective tumor sections). Staining in each section was shown in a merged picture of two photos taken under a Zeiss Axioscope microscope (5×). Scale bar: 400 µm.

**Figure 3 cancers-11-00765-f003:**
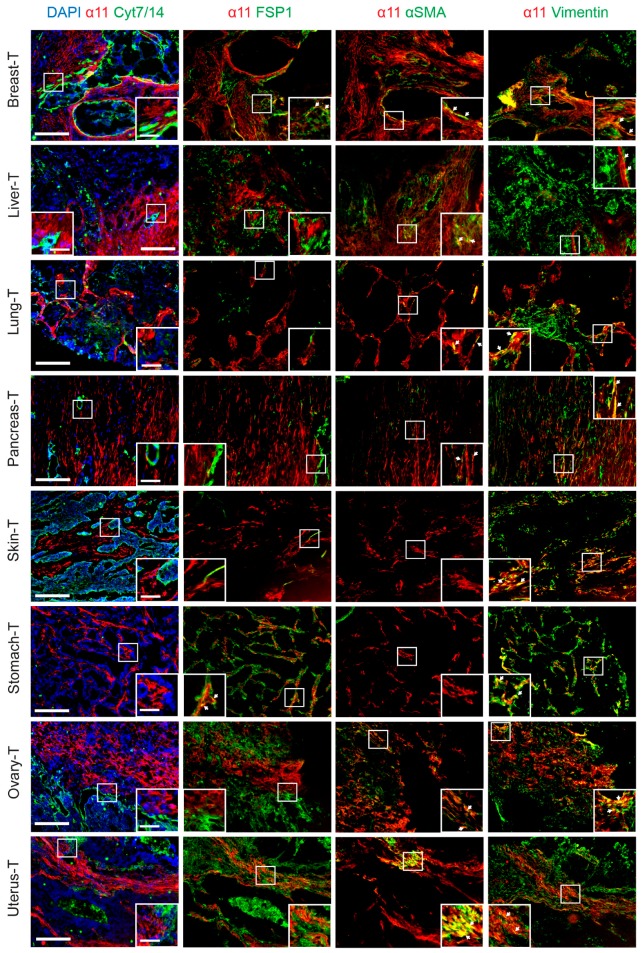
Integrin α11 co-localization with various markers in selected tumor array sections. The sections were stained with α11 203E3 mAb (red,) combined with cytokeratin 7 and 14 (green), FSP1 (green), αSMA (green), and vimentin (green), respectively, as indicated. DAPI (blue) was used for counterstaining only in the combination of integrin α11 and cytokeratin 7/14. Pictures shown were taken under Zeiss Axioscope microscope (10×). Scale bar: 200 µm.A close-up image of a region of interest is inserted in each picture (scale bar: 50 µm). Arrows denote co-localization of integrin α11 with other stroma markers.

**Figure 4 cancers-11-00765-f004:**
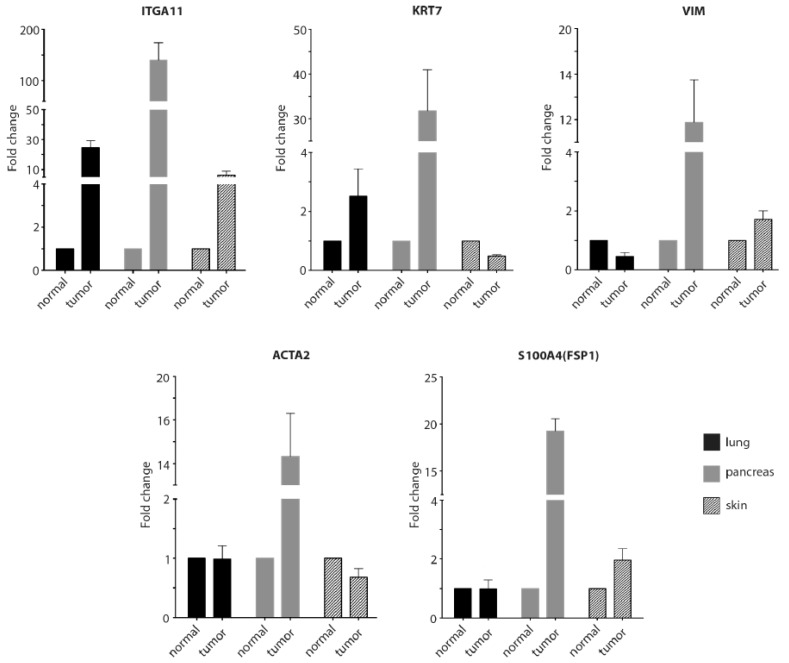
Comparison of mRNA expression of integrin α11 and various markers in selected normal and tumor tissues. Total RNA was extracted from the normal and tumorous lung, pancreas and skin tissues on which immunostaining had previously been performed. mRNA levels of integrin α11 (ITGA11), cytokeratin 7 (KRT7), vimentin (VIM), alpha smooth muscle actin (αSMA) (ACTA2), and fibroblast-specific protein 1 (FSP1) (S100A4) were analyzed by RT-qPCR. Glyceraldehyde-3-phosphate dehydrogenase (GAPDH) was amplified as a reference gene for normalization. Each gene expression level is presented as a fold change in tumor tissue relative to the normal tissue. Shown is the average fold change of the mRNA extracted from each sample, but reverse transcribed and amplified in three independent experiments. Error bar indicates the standard deviation from the average.

**Figure 5 cancers-11-00765-f005:**
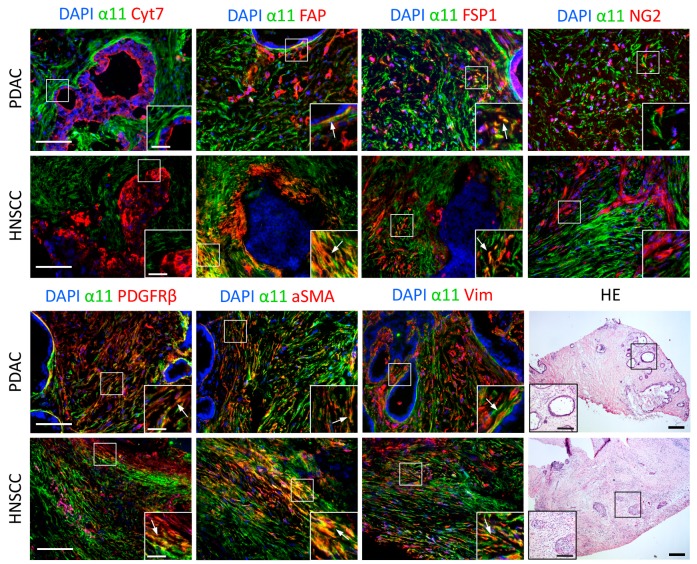
Determination of integrin α11 co-expression with various stroma markers. Fresh-frozen tumor sections from human pancreatic ductal carcinoma (PDAC) and head and neck squamous carcinoma (HNSCC) were co-stained with α11 203E3 mAb (red) and the respective tumor and stroma cell markers, as indicated (green). Cell nuclei were stained with DAPI (blue) and used as counterstaining. The photos were taken under a Zeiss Axioscope microscope (20×). Scale bar: 100 µm. A close-up image of a region of interest is inserted in each picture (scale bar: 25 µm). Arrows denote co-localization of integrin α11 with other stroma markers.HE staining of the sequential section from the PDAC or HNSCC patient was shown in parallel. The photos were taken under a Nikon Eclipse E600 microscope (5×). Scale bar: 200 µm. Inserts show higher magnification of the selected area (scale bar: 100 µm).

**Figure 6 cancers-11-00765-f006:**
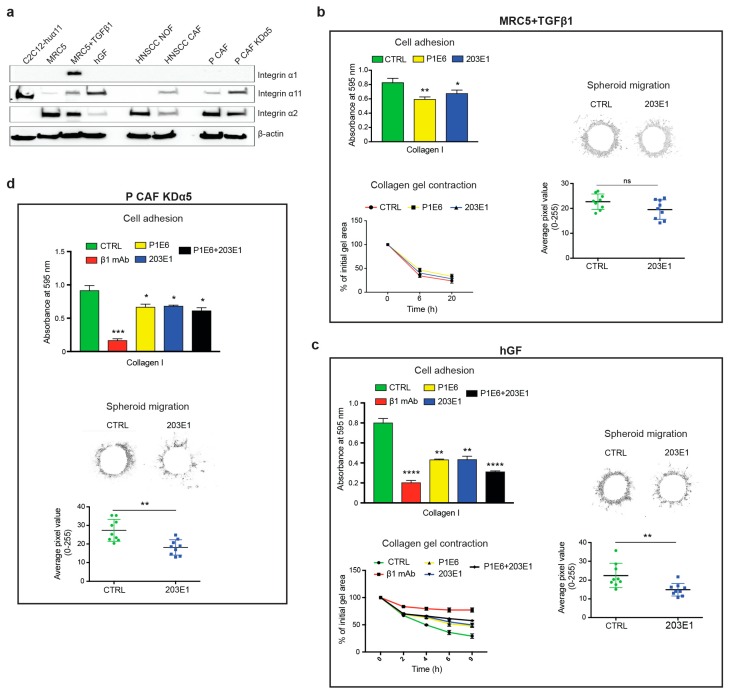
Effect of integrin α11 mAbs on cell-collagen interactions in fibroblasts and cancer-associated fibroblasts. (**a**) Western blot showing the total protein expression of integrin α1, α2, α11, and α2 in; MRC5 fibroblasts with or without TGF-β1 (the MRC5 cells were treated with 5 ng/mL TGF-β1 for 48 h to induce α11 integrin); human gingival fibroblasts (hGF); cancer-associated fibroblasts (CAFs) and normal fibroblasts (NOFs) from a head and neck squamous cell carcinoma (HNSCC) patient; CAFs from a pancreatic adenocarcinoma (pCAF) and pCAFs with integrin α5 knockdown (pCAFKDα5). C2C12-huα11 cells were used as a positive control. (**b**) Effect of integrin antibodies on α11β1-mediated cell adhesion of MRC5 cells. MRC5 cells treated with TGF-β1 were assayed in cell adhesion, collagen gel contraction and spheroid migration in the presence of control antibodies (CTRL), α2 integrin mAb (P1E6) or α11 integrin mAb (203E1). (**c**) Effect of integrin antibodies on α11β1-mediated cell adhesion of human gingival fibroblasts. hGF cells were assayed in cell adhesion, collagen gel contraction and spheroid migration in the presence of control antibodies (CTRL), α2 integrin mAb (P1E6), α11 integrin mAb (203E1), or β1 integrin mAb (mAb 13). (**d**) Effect of integrin antibodies on α11β1-mediated cell adhesion of pCAFKDα5. pCAFKDα5 cells were assayed in cell adhesion, and spheroid migration in the presence of control antibodies (CTRL), α2 integrin mAb (P1E6), α11 integrin mAb (203E1), or β1 integrin mAb (mAb 13). For cell adhesion, the cells were treated with antibodies and allowed to adhere to collagen I in serum free conditions for 50 min. For spheroid data spheroid migration was analyzed after 24 h. Results shown here are representative images of the spheroid after image processing with ImageJ. The radius of the region of interest from each individual spheroid was measured using Radial Profile plugin from ImageJ. Means ± SEM of at least three independent experiments are shown and analyzed with one tailed, unpaired *t*−test * *p* < 0.05, ** *p* < 0.01, *** *p* < 0.001, **** *p* < 0.0001.

**Table 1 cancers-11-00765-t001:** Donor and patient information on the tissue array sections.

Table	Age	Sex	Pathological Diagnosis	Differentiation	TNM or Stage
Brain	70	F	Normal		
Brain Tumor	36	F	Oligodendroglioma	N/A	Stage III
Breast	40	F	Normal		
Breast Tumor	47	F	Invasive Ductal Carcinoma	N/A	T_unknown_N_0_M_0_
Colon	87	F	Normal		
Colon Tumor	70	M	Adenocarcinoma, Mucuous	Moderately	T_2_N_0_M_0_
Skeletal Muscle	79	M	Normal		
Skeletal Muscle Tumor	50	M	Rhabdomyosarcoma	Poorly	T_3_N_0_M_0_
Kidney	44	M	Normal		
Kidney Tumor	37	M	Renal Cell Carcinoma	Moderately	T_3_N_0_M_1_
Liver	64	M	Normal		
Liver Tumor	44	M	Hepatocellular Carcinoma	N/A	T_3_N_0_M_0_
Lung	83	F	Normal		
Lung Tumor	70	M	Adenocarcinoma	Moderately	T_unknown_N_0_M_0_
Pancreas	86	F	Normal		
Pancreas Tumor	53	M	Adenocarcinoma	Poorly	T_unknown_N_0_M_0_
Prostate	50	M	Normal		
Prostate Tumor	66	M	Adenocarcinoma	N/A	Gleason 4 + 3 = 7
Skin	61	F	Normal		
Skin Tumor	48	M	Carcinoma, Sweat Gland	N/A	T_1_N_0_M_0_
Small Intestine	70	F	Normal		
Small Intestine Tumor	68	M	Malignant Mesenchymoma	Well	T_2_N_0_M_1_
Stomach	56	M	Normal		
Stomach Tumor	54	M	Adenocarcinoma, Ulcer	Moderately	T_2_N_0_M_0_
Ovary	37	F	Normal		
Ovary Tumor	54	F	Cystadenocarcinoma, Serous	Poorly	T_2_N_0_M_0_
Uterus	68	F	Normal		
Uterus Tumor	55	F	Adenocarcinoma	Poorly	T_unknown_N_0_M_0_

N/A, not available; M, male; F, female.

**Table 2 cancers-11-00765-t002:** Summary of integrin α11 expression and its co-localization with other stroma markers in tumor sections of the tissue array.

Tumor Tissue	Pathological Diagnosis	α11 Expression in Stroma	Co-Localization α11/FSP1	Co-Localization α11/αSMA	Co-Localization α11/vimentin
Brain	Oligodendroglioma	-			
Breast	Invasive Ductal Carcinoma	+++	+	+	++
Colon	Adenocarcinoma, Mucuous	-			
Skeletal Muscle	Rhabdomyosarcoma	?			
Kidney	Renal Cell Carcinoma	-			
Liver	Hepatocellular Carcinoma	+++	-	++	+
Lung	Adenocarcinoma	++	-	+	++
Pancreas	Adenocarcinoma	+++	-	++	++
Prostate	Adenocarcinoma	-			
Skin	Carcinoma, Sweat Gland	++	-	-	++
Small intestine	Malignant Mesenchymoma	-			
Stomach	Adenocarcinoma, Ulcer	++	++	-	++
Ovary	Cystadenocarcinoma, Serous	+++	-	++	++
Uterus	Adenocarcinoma	+++	-	++	++

-, no expression or co-localization; +, low expression or co-localization; ++, medium expression or co-localization; +++, high expression or co-localization; ?, uncertain expression.

**Table 3 cancers-11-00765-t003:** List of Primers for the quantitative Polymerase Chain Reaction (qPCR).

Human Gene	Forward Primer	Reverse Primer	Product Length
*ITGA11*	5’-GTGGCAATAAGTGGCTGGTC	5’-GACCCTTCCCAGGTTGAGTT	122 bp
*KRT7*	5’-ACTCATGAGCGTGAAGCTGG	5’-ATCACAGAGATATTCACGGCTCC	117 bp
*VIM*	5’-TGGACCAGCTAACCAACGACAAAG	5’-TCCTCTCTCTGAAGCATCTCCTCC	112 bp
*ACTA2*	5’-AGCCAAGCACTGTCAGGAATC	5’-TGTCCCATTCCCACCATCAC	192 bp
*S100A4*	5’-GCAAAGAGGGTGACAAGTTCAAGC	5’-CCTGTTGCTGTCCAAGTTGCTC	137 bp

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
