# Peer review of "α11β1 Integrin is Induced in a Subset of Cancer-Associated Fibroblasts in Desmoplastic Tumor Stroma and Mediates In Vitro Cell Migration"

_cancers, 2019, doi:10.3390/cancers11060765_

Reviewer 1 Report

In this manuscript, the authors described two new monoclonal antibodies against integrin alpha 11 and showed evidence that one of them can be neutralizing antibody to block integrin alpha 11 function. In addition, by performing the immunostaining on tumor tissue array, the correlation of the expression of integrin alpha 11 and other cancer associated fibroblast markers were analyzed. Generally, the data present well support the conclusions.

I have only a few minor comments for the authors to consider when revising the manuscript,

1)      As to show the superiority of the new mAbs, the authors should consider using some commercially available polyclonal antibody against integrin alpha 11 side-by-side in some key experiments, such as the immunostaining on the same type of tissues or consecutive slides, and in vitro cultured cells.   

2)      Figure 3, table 2, figure 5 are all quite descriptive. It would be better to find a way to quantify those images and show a statistical analysis to support authors’ description of the co-localization of integrin alpha 11 and CAF markers.

3)      It would be better to clearly label the cell lines used in Figure 6b, 6c, and 6d.

4)      Figure 6a clearly showed that HNSCC CAFs and PDAC CAFs have a similar level of alpha 11 protein whereas hGF cells have a significantly higher level of alpha 11 protein as compared to PDAC CAFs, which are not consistent with the authors’ descriptions of this data in the manuscript.  Line 272 Expression of α11 was modest in HNSCC CAFs; Line 273, PDAC CAFs expressed high levels of α 11; Line 274 PDAC CAFs expressed comparable levels of α11 as hGF cells.

5)      The writing can be improved.

Author Response

Reviewer #1

We thank the reviewer for his useful comments. Below we discuss the comments and the changes done in the manuscript.

In this manuscript, the authors described two new monoclonal antibodies against integrin alpha 11 and showed evidence that one of them can be neutralizing antibody to block integrin alpha 11 function. In addition, by performing the immunostaining on tumor tissue array, the correlation of the expression of integrin alpha 11 and other cancer associated fibroblast markers were analyzed. Generally, the data present well support the conclusions.

I have only a few minor comments for the authors to consider when revising the manuscript,

1) As to show the superiority of the new mAbs, the authors should consider using some commercially available polyclonal antibody against integrin alpha 11 side-by-side in some key experiments, such as the immunostaining on the same type of tissues or consecutive slides, and in vitro cultured cells.

Response: We have a long experience with polyclonal a11 antibodies, both our own and commercial ones. One common denominator is that they all stain both adult tissues and tumor tissues non-specifically. We believe that most published data obtained with commercial a11 antibodies are artifactual.

2) Figure 3, table 2, figure 5 are all quite descriptive. It would be better to find a way to quantify those images and show a statistical analysis to support authors’ description of the co-localization of integrin alpha 11 and CAF markers.

Response: We agree that the figures are descriptive. As discussed above, published data produced using commercial antibodies suffer from a non-specificity problem, and protein atlas data obtained with polyclonal a11 antibodies also result from non-specific signals in the majority of tissues. We consider the availability of these new reagents to represent a big step forward in our field of collagen receptors.

The reason for not doing a more quantitative study is that the tumor sections used in the tissue array arer very small (so that it is also impossible to do Western blotting). Also, the great heterogeneity within the tumors would make such quantifications less meaningful.

3) It would be better to clearly label the cell lines used in Figure 6b, 6c, and 6d.

Response: We thank the reviewer for this comment. The labeling has been changed.

4) Figure 6a clearly showed that HNSCC CAFs and PDAC CAFs have a similar level of alpha 11 protein whereas hGF cells have a significantly higher level of alpha 11 protein as compared to PDAC CAFs, which are not consistent with the authors’ descriptions of this data in the manuscript.  Line 272 Expression of α11 was modest in HNSCC CAFs; Line 273, PDAC CAFs expressed high levels of α 11; Line 274 PDAC CAFs expressed comparable levels of α11 as hGF cells.

Response: We apologize for this confusion.

1. Two different types of PDAC CAF were used: control PDAC CAFs (P CAF) and PDAC CAFs in which the integrin a5 has been knocked down (P CAF KDα5). Since the a5 KD CAFs expressed the highest level of a11, we decided to use these in the functional blocking experiments.

2. The control PDAC CAFs expressed modest levels of 11, whereas the PDAC CAFs with 5 knock-down expressed levels comparable to those found in hGF.

Two results from these studies are intriguing:

Firstly, interestingly – and as noted by the reviewer, although hGF and P CAF KDα5 expressed similar levels of α11, the effect of 203E1 was greatest in the hGF cells, a fact that was probably related to the lower levels of α2 integrin in these cells.

Secondly, although α2β1 and α11β1 are the major receptors for fibrillar collagens, the combined effect of blocking them both does not match the effect of blocking the β chain.

There are 12 α-chains associated with β1, so the additional effect is most likely to come from some other integrin contributing to cell-collagen interactions. Although cell attachment is a short-term assay, residual fibronectin could bind to collagen. In the more long-term collagen gel contraction and spheroid assays fibronectin is present in the serum and is also contributed by synthesis. Integrins α5β1 and αvβ1 are candidates for mediating binding to collagen.

We have clarified this issue and moved some parts of text to discussion.

5) The writing can be improved.

Response: The manuscript has now been subjected to professional language editing.

Reviewer 2 Report

In this manuscript the authors describe a further screening of integrin alpha-11 expression in different tumor tissues.  The expression of this receptor in several tissues had  been previously reported by the same research group and other researchers and the main difference reported herein is the introduction of the newly developed monoclonal antibodies to human α11, mAb 203E3 and mAb203E1. being the two mAb covered by patent, details on this two tools are not reported. At line 273 and line 594, authors report that these antibodies are selective for this specific subclass of integrins, but unfortunately no data are reported in support. 

Concerning paragraph 2.3 and figure 4 where mRNA expression of integrin alpha-11 and different markers in selected normal and tumor tissues are reported, the strong increase of mRNA expression for all the markers in pancreatic tumor tissue is justified by the authors by the  agreement with immunohistochemical data, but some further explanations should be added.

Coexpression with other important markers has also been investigated. The main novelty is represented by the study on the alpha-11 expression in a subset of non-pericyte-derived CAFs and the effect in collagen remodeling and CAF migration in the TME. Anyway, the cell adhesion data on these CAFs from pancreatic adenocarcinoma (pCAF) in the presence of alpha-11 mAb are similar to those obtained with alpha-2 mAb and inhibition is  much lower than that observed in the presence of beta-1 mAB.

The data are correct and the paper is written, even if in some paragraph it's quite difficult for the reader and some paragraph could be clarified. Anyway, the conclusions are not very clear and the data reported are not sufficient for the publication on this journal at this stage. In my opinion the manuscript as it is, could be suitable for a more specific journal in the field.

Author Response

Reviewer #2

We thank the reviewer for his useful comments. Below we discuss the comments and the changes done in the manuscript.

In this manuscript the authors describe a further screening of integrin alpha-11 expression in different tumor tissues.  The expression of this receptor in several tissues had  been previously reported by the same research group and other researchers and the main difference reported herein is the introduction of the newly developed monoclonal antibodies to human α11, mAb 203E3 and mAb203E1 being the two mAb covered by patent, details on this two tools are not reported.

1. At line 273 and line 594, authors report that these antibodies are selective for this specific subclass of integrins, but unfortunately no data are reported in support.

Response: As mentioned in Material and Methods the screening step involved screening hybridomas against C2C12 cells, which express a variety of mouse integrins. a11 mAbs do not react with these cells excluding cross-reactivity with other mouse integrins.

Hybridomas were also screened for reactivity with A431 cells, which express a variety of human integrins, including a2b1, a3b1, a5b1, avb5 and avb6 [1]. The a11 mAbs do not react with these cells, excluding cross-reactivity with the human b1 integrin chain av integrin chains or the multiple additional integrin a chains expressed by A431 cells.

The integrin a11 has a distinct molecular weight from the other integrin a-chains. Cross-reactivity with other integrin chains would be detected in immunoprecipitation and Western blotting that we do not observe.

The prime suspect for cross-reactivity is the closest relative of a11 in fibroblasts, the a2 integrin (35% sequence identity). All the hybridoma clones were tested against a2b1 protein in the process of screening with lack of reactivity, which is in agreement with lack of reactivity of a11 mAbs with the α2 expressing A431 cells. We have now better described the specificity and clarified the lack of cross-reactivity in the manuscript.

2. Concerning paragraph 2.3 and figure 4 where mRNA expression of integrin alpha-11 and different markers in selected normal and tumor tissues are reported, the strong increase of mRNA expression for all the markers in pancreatic tumor tissue is justified by the authors by the  agreement with immunohistochemical data, but some further explanations should be added.

Response: We have changed the writing to clarify this point: "The qRT-PCR data demonstrated increased RNA levels of integrin α11 (ITGA11) in the lung, pancreas and skin tumor tissue relative to the normal tissues, with the greatest increase in α11 RNA to be found in the pancreas tumor. Interestingly, vimentin (VIM) and αSMA (ACTA2) that we showed to both co-localize with integrin α11 in the pancreatic cancer tumors, displayed also increased expression in this tumor tissue."

3. Coexpression with other important markers has also been investigated. The main novelty is represented by the study on the alpha-11 expression in a subset of non-pericyte-derived CAFs and the effect in collagen remodeling and CAF migration in the TME. Anyway, the cell adhesion data on these CAFs from pancreatic adenocarcinoma (pCAF) in the presence of alpha-11 mAb are similar to those obtained with alpha-2 mAb and inhibition is  much lower than that observed in the presence of beta-1 mAB.

Response: We thank the reviewer for pointing this out. There are 12 α-chains associated with β1, so that the additional effect is most likely to come from some other integrins contributing to cell-collagen interactions. Although cell attachment is a short-term assay, residual fibronectin could bind to collagen. In the more long-term collagen gel contraction- and spheroid assays fibronectin is present in serum and is also contributed by synthesis. α5β1 and integrin αvβ1 are candidates for mediating binding to collagen.

We have now added a sentence in the discussion to clarify this point and a recent reference supporting this indirect mode of interaction with collagen that is also described in detail in a recent review of ours [2,3].

4. The data are correct and the paper is written, even if in some paragraph it's quite difficult for the reader and some paragraph could be clarified.

Response: We have rewritten some paragraphs.

References:

1.         Pidgeon, G.P.; Tang, K.; Cai, Y.L.; Piasentin, E.; Honn, K.V. Overexpression of platelet-type 12-lipoxygenase promotes tumor cell survival by enhancing alpha(v)beta(3) and alpha(v)beta(5) integrin expression. Cancer Res 2003, 63, 4258-4267.

2.         Miyazaki, K.; Oyanagi, J.; Hoshino, D.; Togo, S.; Kumagai, H.; Miyagi, Y. Cancer cell migration on elongate protrusions of fibroblasts in collagen matrix. Sci Rep 2019, 9, 292, doi:10.1038/s41598-018-36646-z.

3.         Zeltz, C.; Orgel, J.; Gullberg, D. Molecular composition and function of integrin-based collagen glues-introducing COLINBRIs. Biochim Biophys Acta 2014, 1840, 2533-2548, doi:10.1016/j.bbagen.2013.12.022.

Reviewer 3 Report

see at the attachment

Author Response

Reviewer #3

We thank the reviewer for his useful comments. Below we discuss the comments and the changes done in the manuscript (see the PDF file for the figures).

In this manuscript, the authors developed and characterized two new mAbs against

integrin-a11, one works for immunofluorescence (203E3) and another exhibits function

blocking properties. Using moAb 203E3 on a tissues array containing 14 cancers for the

immunohistochemistry, they characterize integrin-a11 expression in tumor stromal

component from various cancer types with varied expression levels and patterns. The

function block moAb 203E1 was able to inhibit fibroblasts-collagen interactions, that

supports integrin-a11 's pro-invasive role in CAFs.

This work is important to expand our knowledge of the role of the unique collagen

receptor, integrin- a 11 in tumor stromal CAFs for promoting cancer progression, with

these novel moAbs being precious tools to study in a broad range of cancer types.

However, the current manuscript is in a pretty preliminary shape; the paragraph in the

introduction and discussion are relatively insufficient to make point clear, and there are

numerous errors and inadequate sentences that need to be intensively fixed.

1. Do the authors mean to demonstrate that integrin- a 11 is expressed in a some subset

of CAFs based on distinct co-staining pattern with a few markers? The characterization

of CAFs by just a limited markers and only 1 sample case from one cancer type is not

very conclusive.

2. The statement of all fibroblast markers used (FAP, aSMA, FSP1, NG2,

vimentin, and PDGFR) needs to be better clarified for what type of stromal cells

represented in the tumor microenvironment.

3. Similarly, with the data from the function

blocking assays, it is not convincing to say the effect of the moAbs is

expression-dependent with limited number of cell types.

Rather, it should be more emphasized in the manuscript that the study confirmed

integrin-a 11 as an marker for CAFs and extended integrin- a 11 's involvement in the

broader range of cancer types, and the newly developed moAbs are precious tools for

further investigations.

Here's the list of point to be addressed.

1• What are the epitope sequences recognized by mAb 203E1 and mAb203E3? It is

important to map especially where the function blocking mAb203E1 binds to.

Response: The mapping of the epitope sequences recognized by mAb 203E1 and 203E3 is ongoing studies. We have generated chimeric molecules in which the a11 I domain has been replaced with the a2 I domain. Preliminary data demonstrate that the epitope for the function blocking antibody is not in the a11 I domain. Various deletion constructs are currently expressed in HEK 293 cells for further epitope mapping.

2• Figure 1 b - A negative control, such as non-immunized serum and class matched

immunoglobulin is required for immunoprecipitation. The loading of the cell lysis

used for immunoprecipitation should be shown as well to show the

immunoprecipitation efficiency.

Response: No pre-immune control was performed in this instance since, as specified in Materials and Methods, the sample was pre-cleared with pre-immune serum.

We have now performed an immunoprecipitation with a loading control (input) and with a matched isotope (mouse IgG1) to demonstrate the lack of non-specific reactivity under the given conditions. The result is presented in the Figure 1b.

3• Figure 2 - its strange that Sk. Muscle rhabdomyosarcoma is positive for

cytokeratins as it's a mesenchymal neoplasia.

Response: We actually regard this weak positivity as background staining (as compared to other tissues). Two notorious tissues for background staining are skin and muscle. We have chosen not to comment on this further.

4• Integrin-a11 staining in lung tumorlooks as clear as those of breast, liver, pancreas and ovary, yet it is considered asweak (in line-159).

Response: The reviewer is correct and the text has been modified.

5• The HE staining of the normal and tumor tissue histology sections should be

shown.

Response: We have now performed H&E staining on HNSSC and PDAC tissues and have added the new data in the Figure 5.

We have also stained normal and tumor tissues from tissue array for H&E (see Figure 1 below), however from a different lot that the one we used in our study. Therefore, we would like not to add the H&E stained section in the manuscript.

Figure 1. H&E staining in array sections from normal and tumor adult human tissues. Scale bar: 400 µm.

6• section 2.51 Figure 5 - FAP, FSP, NG2, PDGFR, a SMA and vimentin are the

widely used markers for CAFs. Yet, the purpose of the staining is not clearly

described in this section. First, integrin- a 11 is co-localized with the common

CAF-markers.

Response: The issue of fibroblast heterogeneity is complex and our immunostainings can be seen as a first attempt to better characterize cells expressing integrin α11. FSP-1 is widely recognized in immune cells in some tissues, NG2 is a pericyte marker and PDGFβR is a stromal marker recently shown by us to form a complex with a11b1 in breast cancer tissue [1]. For detailed results, every tissue will require extensive studies. We have therefore chosen not to go into detail about the interpretations obtainable from the different tissues. This will be the focus of future studies. Some additional text has been added to the manuscript.

Minor collections (some but not all)

The manuscript has now been subjected to professional language editing.

Line 41 -- The extracellular matrix (ECM) in addition to serving as a structural - The

extracellular matrix (ECM) is serving as a structural scaffold in addition to a

reservoir of growth factors and cytokines.

Response: We have corrected the sentence.

Line 45 - The major cell types in the tumor stroma of solid tumors include cancer-associated fibroblasts (CAFs) of varying origin, endothelial cells, pericytes, mesenchymal stem cells and immune cells [3,4]. CAF is a major cell type within the stroma contributing to ECM synthesis and ECM remodeling, but CAFs also take part in paracrine signaling that affect growth and invasive properties of the tumor cells, chemoresistance as well as taking part in establishing metastatic niches [3-5].

Response: We have modified the text according to the reviewer's comment.

Line 49 - Importantly, subsets of CAFs have been recognized and a specific subset of

myofibroblastic CAFs (myo-CAFs) has been implicated in the production of collagen

[6].

Line 59 - The balance between cells of different origins is dynamic ~ The balance among

Cells

Response: We have made the changes accordingly.

Line 60 -In tissue fibrosis, genetic-method-based cell linage tracing and more careful use of antibodies ~ more careful than what?

Response: We have replaced "careful" by "stringent".

Line 77- The contribution of EMT or partial EMT to tumor-stroma interactions appears to be especially important in contributing an invasive mesenchymal tumor cell type and in creating niches for cancer stem cell formation [22].

Response: We have now modified the original text.

Line 86 - A detailed in vitro study using spheroids, and which in-depth studied the process of invasion, ? unclear sentence

Response: We have change the sentence to "A detailed in vitro study using breast cancer cell spheroids".

Line 106 -- supernatants from hybridoma clones were further characterized after

proper subcloning 7 hybridoma clones were further subcloned? Need a correction or explanation.

Response: We have rewritten the whole paragraph. This sentence is not in the manuscript anymore.

line 132 -- collagen I using and immunostained using mAbs 203El and 203E3 ~

remove

Response: We have rewritten the sentence.

line 133 - scale bar: ~ 720 ~

Line 134 -- incubated with beta-l mAb ~ add

Response: We have made the appropriate corrections.

Line-l65 - 4 tissues consistently lacked specific integrin all signal in the tissue array

analyzed, 7 only one tissue sample for each tumor type is tested, so the result

can not be considered as ' consistently'

Response: We have removed "consistently".

Line-232 - integrin all is also expressed in neural crest-derived fibroblastic cells in the

mouse head region ~ what is this region? Brain?

Response: During the rewriting process, we have removed this reference.

Line-235 - squamous carcinoma (HNSCC) as as an additional tumour source for the added characterization of ~ additional

Response: During the rewriting process, we have removed this sentence.

Line-261 - MRC5 fibroblasts ~ should add a brief description of MRC5; "human lung

fibroblasts"

Response: We have now better described the MRC5 cell line: "Human lung embryonic MRC5 fibroblasts".

Line-264 - at our level of detection ~ what does it mean? Its confusing and unnecessary.

Response: We have removed this confusing part.

Line-266, 269-was ~ were

Response: We have corrected this mistake.

Line-273 - However, PDAC CAFs [39] expressed high levels of ----.ll. ~ this citation is not clear to be here.

Response: We have removed the citation from this paragraph.

Line-339 - an in vitro spheroid model for breast cancer metastasis ~ invasion, human MDA tumor cells ~ what is MDA tumor cells?

Response: We have change the text to " that human breast cancer tumor cells at the invasive front in an in vitro spheroid metastasis model".

Line-329, line 355 - the sentences use incorrectly repetitive words-

--The finding that bright a II immunostaining is found

--The newly raised monoclonal antibodies characterized in this study, are raised….

Response: We have modified the sentences accordingly.

References:

1.         Irina Primac; Erik Maquoi; Silvia Blacher; Ritva Heljasvaara; Jan Van Deun; Hilde YH Smeland; Canale, A.; Thomas Louis; Nor Eddine Sounni; Christel Pequeux, et al. Stromal integrin α11 triggers PDGF-receptor-β signaling to promote breast cancer progression. Under review  J. Clin. Invest. 2019.

Reviewer 4 Report

The manuscript entitled “α11β1 integrin is induced in a subset of cancer-associated fibroblasts in desmoplastic tumor stroma and mediates cell migration” by Zeltz C. et al. reports that integrin α11β1, expressed in a subset of non-pericyte-derived Cancer-Associated Fibroblasts, constitutes an important receptor for collagen remodeling involved in CAF migration in the TME. In my opinion, the Author’s contribution to the knowledge of the role of integrin in cancers is extremely interesting.

Although the experimental procedures used are extremely appropriate for the purposes of the research, unfortunately, the same attention to detail has not been applied to the manuscript writing, thus I suggest a careful revision of the manuscript.

The Authors evaluated the expression of integrin α11β1 in different populations of intratumoral fibroblasts, using two specific monoclonal antibodies, which, according to the Author’s results, recognize the heterodimeric receptor. Despite this, frequently, throughout the manuscript, the Authors refer only to α11 subunit. It is extremely necessary to clarify this point.

In addition:

- Title, use "in vitro cell migration".

- The use of “C2C12-hu α11” expression instead of “C2C12-α11”, as it is more accurate.

- Please revise lines 230-236, not clear.

- Line 339, please explain which MDA cells have been used.

- Lines 552, please explain why the Authors use PBS WITH Ca2+ and Mg2+ to remove non-adherent cells.

- Lines 237, 362 and 448, something is missing.

Despite the need for a language detailed revision, the results are extremely intriguing.

Author Response

Reviewer #4

We thank the reviewer for his useful comments. Below we discuss the comments and the changes done in the manuscript.

The manuscript entitled “α11β1 integrin is induced in a subset of cancer-associated fibroblasts in desmoplastic tumor stroma and mediates cell migration” by Zeltz C. et al. reports that integrin α11β1, expressed in a subset of non-pericyte-derived Cancer-Associated Fibroblasts, constitutes an important receptor for collagen remodeling involved in CAF migration in the TME. In my opinion, the Author’s contribution to the knowledge of the role of integrin in cancers is extremely interesting.

Although the experimental procedures used are extremely appropriate for the purposes of the research, unfortunately, the same attention to detail has not been applied to the manuscript writing, thus I suggest a careful revision of the manuscript.

We thank the reviewer for his input.

The Authors evaluated the expression of integrin α11β1 in different populations of intratumoral fibroblasts, using two specific monoclonal antibodies, which, according to the Author’s results, recognize the heterodimeric receptor. Despite this, frequently, throughout the manuscript, the Authors refer only to α11 subunit. It is extremely necessary to clarify this point.

Response: We apologize for this confusion. The functional receptor is the integrin heterodimer. The individual chains were formerly correctly labeled as the integrin a11 subunit and the integrin b1 subunit. In the field of integrins one often drops the word ‘subunit’ and says just ‘integrin a11’ and ‘integrin b1’, which, as noted by the reviewer, is in formal terms wrong, since integrin is only functional as a dimer. However, the antibodies that we have generated are specific to the integrin a11 subunit.

We have gone through the manuscript and tried to clarify this point at the most central places throughout.

In addition:

- Title, use "in vitro cell migration".

Response: We have changed the title accordingly.

- The use of “C2C12-hu α11” expression instead of “C2C12-α11”, as it is more accurate.

Response: We agree with the reviewer, and we have made this modification in the manuscript.

- Please revise lines 230-236, not clear.

We have previously shown that integrin α11 is also expressed in neural crest-derived fibroblastic cells in the mouse head region but we have also reported on an α11 up-regulation in desmoplastic squamous carcinoma stroma of head and neck using an α11 polyclonal antibody. Based on this data, we chose head and neck squamous carcinoma (HNSCC) as as an additional tumour source for the added characterization of a11-expression in assorted CAFs, using the novel a11-reactive mAbs.

Response: During the rewriting process, we have removed this paragraph.

- Line 339, please explain which MDA cells have been used.

Response: We have change the text to " that human breast cancer tumor cells at the invasive front in an in vitro spheroid metastasis model".

- Lines 552, please explain why the Authors use PBS WITH Ca2+ and Mg2+ to remove non-adherent cells.

Response: This is standard protocol in the integrin field in order to support integrin-mediated adhesion. Integrins are dependent on divalent cations to function (both to keep the correct conformation of the integrin alpha subunit and for some integrins to take part directly in ligand binding). This applies especially to collagen-binding integrins. Crystallization of a2 integrin in a complex with GFPGER binding shows direct Mg2+ bridging between the MIDAS (metal ion dependent adhesion site) of integrin and the GFPGER site in collagen[1].

- Lines 237, 362 and 448, something is missing.

237 For this purpose six different stroma markers were chosen for co-staining with α11 mAb 203E3.

Response: We have now rewritten these parts.

Despite the need for a language detailed revision, the results are extremely intriguing.

Response: The manuscript has now been subjected to professional language editing. We thank the reviewer for finding our contribution to cancers extremely interesting. We agree with the reviewer. As a confirmation that a11 has a unique function in the tumor stroma, our recent study in Itga11-/-//PyMT mice demonstrates an important role for integrin a11 in the breast cancer model.

References:

1.         Emsley, J.; Knight, C.G.; Farndale, R.W.; Barnes, M.J.; Liddington, R.C. Structural basis of collagen recognition by integrin alpha2beta1. Cell 2000, 101, 47-56.

Round  2

Reviewer 2 Report

The revised manuscript is now suitable for publication

Author Response

We are glad that the reviewer now finds MS suitable for publication.

Reviewer 3 Report

The authors made a remarkable improvement in the description and provide satisfactory data on current manuscript.

 The abbreviations needs to be clarified,

e.g., line 50 and 63 " myCAFs", line 69" pyMT"

There are some simple errors

line 145

line 240

line 296 (when involved?)

line 315

line 603

Author Response

Thank you for pointing these thigns out.

-line 50 and 63: myCAFs is  abbreviation for myofibroblastic cancer-associated fibroblasts, CAF abbreviation is already spelled out on line 45.

line 69:the transgenic polyoma middle T oncogene (PyMT)  has been spelled out

line 144: mAb 203E3 a11 has been changed to α11 mAb 203E3 

line 240: repetiotion of "to be" removed

line 296: "when involved" removed

line 315: "expressed" changed into detected

line 614: symbol in α11 corrected